# The Relationship between Meaning in Life and the Childhood Family Environment among Emerging Adults

**DOI:** 10.3390/ijerph20115945

**Published:** 2023-05-24

**Authors:** Emily Dameron, Marcie C. Goeke-Morey

**Affiliations:** Department of Psychology, The Catholic University of America, Washington, DC 20064, USA; goekemorey@cua.edu

**Keywords:** meaning in life, childhood family environment, loneliness, attachment

## Abstract

This study investigates the impact of the childhood family environment on reported meaning in life among emerging adults (*n* = 507) at a private, urban, religious university. This study found that participants who reported growing up in an emotionally warm family environment ultimately reported more meaning in life as adults and that this effect was mediated by loneliness. This suggests that people from emotionally cold and rejecting early family environments may struggle with meaning in life as adults because they are lonelier. This research contributes a developmental perspective to understanding meaning in life. The public health implications of these findings are discussed. Future research should consider accounting for the effects of early life experiences on meaning in life.

## 1. Relationship between Meaning in Life and the Childhood Family Environment in Emerging Adults

Over the last two decades, a growing body of research has emerged on the construct of meaning in life [1,2], but very few studies have investigated meaning in life from a developmental perspective [3]. How might childhood factors, such as the childhood family environment, impact whether one has a sense of meaning in life in adulthood? The childhood family environment influences the mental and physical health of children as they grow into adulthood. Research shows that growing up in an emotionally cold and risky family environment predicts mental and physical illness in adulthood [4,5,6], while growing up in a family environment defined by warmth, security, and safety predicts positive mental and physical health outcomes [7,8].

The existing research on the link between the childhood family environment and mental health in adulthood often conceptualizes mental health as the absence of negative symptoms or psychopathology [9]. However, research indicates that the absence of the negativity does not necessarily produce positive outcomes, such as flourishing and positive functioning, e.g., [7,10]. Thus, there is very little research examining the link between childhood factors and positive mental health factors. As Whitaker and colleagues pointed out [9], only six known studies have examined the impact of parental warmth in childhood on wellbeing in adulthood (seven studies after including the study by Whitaker and colleagues [9]). However, none of these studies have specifically investigated meaning in life as an outcome variable.

In addition, much of the existing body of research has examined the link between meaning and health among midlife and older adults, e.g., [10,11,12]. No known studies have specifically investigated the link between parenting received in childhood and meaning among emerging adults. Emerging adulthood is an important period of psychosocial development for several reasons [13]. First, emerging adulthood represents the first time, for many, of being away from the childhood family environment for extended periods of time. Second, psychologists have argued that healthy development in emerging adulthood requires the exploration and definition of selfhood and values and the cultivation of meaning and purpose [13,14,15]. According to Damon [16], a key indicator of positive and healthy development among emerging adults is cultivating a purpose and possessing a sense of meaning in life. However, as Hill and colleagues pointed out [3], people who experienced adversity in childhood and adolescence have more difficulty with these tasks.

This study seeks to fill these gaps by investigating how a person’s childhood family environment influences that person’s sense of meaning in life in emerging adulthood. Specifically, this study asks: Does the emotional warmth, security, and safety that children experience in their relationships to their parents and within the household impact the degree to which they believe their lives are meaningful as emerging adults? This study seeks to fill an additional gap. Though much research has linked parenting and adult mental health, few studies have investigated why such a link exists; is there a mediator that may explain why the childhood family environment impacts their sense of meaning in life in emerging adulthood?

One potential mediator between the childhood family environment and meaning in life in emerging adulthood is perceived loneliness. The ability to form healthy and satisfying intimate relationships in adulthood is greatly influenced by the childhood family environment, which includes the emotional quality of the parenting received by the child and whether the household environment was risky (e.g., the presence of violence, drug or alcohol abuse, and general chaos) [6,17]. As emerging adults leave their families of origin and enter new social worlds, however, they struggle with this task and suffer from increasing rates of loneliness [18]. These effects have been seen across the population of emerging adults, but they are especially acute for emerging adults who grew up in households defined by emotional abuse or neglect [6,17]. Research indicates that children who grow up in homes defined by cold parenting, abuse, and dysfunction become emerging adults who struggle to form intimate attachments to others and experience more loneliness [17]. Researchers have also found that loneliness predicts lower levels of meaning in life [19].

### 1.1. Childhood Family Environment

The emotional warmth, security, and safety that children experience in their relationships with their parents impact the quality of the relationships they have later in life and the degree to which adults feel lonely or connected to those around them [20]. Thus, the childhood family environment may play a role in how meaningful an emerging adult believes their life to be. Though there are now thousands of studies on the childhood family environment, there is very little research investigating the connection between the childhood family environment and meaning in life.

The study of the childhood family environment originated with attachment theory with the work of John Bowlby and Mary Ainsworth [21,22]. Bowlby and Ainsworth’s work revealed that children need warm, caring, stable relationships to a primary caregiver, such as a parent, to thrive emotionally and physically [20,23]. According to Hazan and Shaver [24], attachment styles formed in childhood carry into one’s later relationships. Securely attached children become adults who trust others and are comfortable with intimacy and vulnerability [24], while insecurely attached children become adults who struggle in their relationships. They have difficulties trusting and getting close with others. Relationships characterized by secure attachment are ultimately more satisfying than ones characterized by insecure attachment [20,25]. Importantly, there is a strong connection between adolescent attachment patterns and loneliness [26]. For example, adolescents and young adults who perceived their parents as caring were less likely to report feelings of loneliness and feel more social support [27,28,29]. Other longitudinal and cross-sectional research consistently reveals that adults who are insecurely attached report more feelings of loneliness [26]. As Mikuliner and Shaver [30] write, summarizing the research connecting insecure attachment in adults and loneliness, “patterns of chronic loneliness were similar in certain respects to the insecure infant attachment patterns identified by Ainsworth and her colleagues” (p. 164).

The health of the childhood family environment may be determined by assessing the degree of parental emotional warmth that children experience [31] and the riskiness of the home environment [32]. Parental Acceptance–Rejection Theory [33] is an evidence-based theory of psychosocial adjustment that holds that children inherently seek warmth from and closeness to their parents and that when they are deprived of these needs, they experience worse psychological adjustment (e.g., lower self-esteem and emotional stability, etc.). According to Parental Acceptance–Rejection Theory, parental warmth exists on a spectrum from acceptance (e.g., lovingness, support, affection, care, etc.) to rejection (e.g., coldness, hostility, abuse, absence or withdrawal of love, etc.) [33]. In a meta-analysis, Khaleque [34] found that parental warmth predicted psychological adjustment in children with moderate to large effect sizes. Longitudinal research has shown that low levels of parental warmth in childhood are associated with depression in adulthood [5], feelings of worthlessness [35], mental health problems [36,37], and physical health problems [38]. At the same time, research has found that parental warmth is associated with better mental and physical health in adulthood [7]. The riskiness of the childhood family environment is another important predictor of psychosocial adjustment [4,6,32]. Children exposed to risky factors in the household, such as violence, drug or alcohol abuse, interparental conflict, etc., are at risk for poor psychosocial outcomes [4,6]

The reviewed research shows that the childhood family environment may influence people’s relationships, their health, and their psychological adjustment later in life. In particular, people from emotionally warm, safe, and secure family environments experience less loneliness and have adult relationships defined by a sense of trust, intimacy, and belonging, but children from emotionally cold and risky family environments are lonelier. Given that the childhood family environment influences adult relationships and that adult relationships are important drivers of meaning in life, does the childhood family environment also impact meaning?

### 1.2. Meaning in Life

A key element of meaningful living is connection [39,40]. Steger and colleagues conceptualized meaning as having three components: significance, purpose, and coherence [1]. Purpose, a highly studied component of meaning, is defined as having goals, intentions, and projects that are valued by the self and contribute to society [10,11,12,16,41]. Empirical research of recent years shows that having a sense of meaning and purpose in life comes with many benefits. People who have a sense of meaning in life are ultimately more psychologically resilient when presented with difficult information [42]; their bodies are more physically resilient to illnesses and physical symptomology [41]; they perform better academically [43]; and they are more productive and engaged with their work [44]. Additionally, they experience greater longevity [12,45]. On the other hand, lacking a sense of meaning and purpose in life is associated with many negative outcomes, such as psychopathology and poor physical health, e.g., [10,16,45].

A key finding of the research on meaning is that relationships defined by a sense of belonging are an important source of meaning in life [46]. Belonging is defined by having positive relationships in which one feels accepted [47,48]. In a classic paper, Baumeister and Leary [48] synthesized decades of empirical research to argue that human beings have a need to belong that is driven, in part, by our evolutionary origins. People feel a sense of belonging, according to Baumeister and Leary, when they have frequent positive interactions with others that are based on mutual care. In research, participants rate their relationships to others as their most important source of meaning in life [49,50,51]. According to research by Lambert and colleagues [47], the active ingredient of meaningful relationships is belonging. In one experiment, for example, participants were primed with a sense of belonging, social value, or social support, and those primed with a sense of belonging rated their lives as more meaningful [47].

It is possible to be in close relationships that lack a sense of belonging [47]. In such relationships, there may be frequent contact, but the quality of the connection may be defined by coldness and rejection. Psychologists have shown that rejection is a threat to meaning [52,53,54,55]. In one experiment [55], participants who were made to feel rejected and left out of a social encounter were significantly more likely to say that life in general was meaningless. One proposed mechanism for this phenomenon is loneliness [19]. Experiences of rejection may make people feel alone, and this sense of loneliness may threaten the meaningfulness with which they judge their lives [19]. In support of this proposition, Stillman and colleagues [19] conducted two studies showing that people who have chronic feelings of loneliness rate their lives as less meaningful. Thus, feelings of loneliness, isolation, and rejection may threaten one’s sense of meaning in life.

Given the importance of relationships to meaningfulness, it may be that the quality of early life relationships, specifically within the family context, influences a person’s sense of meaning in life in adulthood. An emotionally warm, safe, and secure family environment provides children with a sense of belonging, i.e., the sense that they are cared for, respected, and loved, and feeling a sense of belonging makes people rate their lives as more meaningful in adulthood. However, if one grew up in a family environment where love was withheld or one did not feel accepted by one’s caregivers, then one arguably may not have felt a sense of belonging within one’s family. The research summarized here suggests that such a lack of belonging may compromise one’s sense of meaning in life. The question is whether that lack of belonging in childhood leaves a lasting impression later in life, such that one continues to feel a lack of belonging in relationships in emerging adulthood.

### 1.3. Meaning and the Childhood Family Environment

There are few studies that link the childhood family environment with meaning in life in emerging adulthood. Homang and Jong [56] examined the link between the childhood family environment and health outcomes and found purpose to be a mediator of the effect. However, they looked at the purpose dimension of meaning alone, rather than meaning as a whole, which includes one’s sense of purpose, coherence, and significance. In addition, their sample consisted of middle-aged adults rather than emerging adults. Hill and colleagues [3] also investigated the link between childhood adversity and purpose, finding that childhood adversity predicts a lessened sense of purpose. However, they, too, studied middle-aged adults and assessed purpose alone, rather than meaning as a whole. Whitaker and colleagues [9] examined whether parental connection might moderate the connection between adverse childhood experiences and psychological wellbeing, as conceptualized by Ryff [57], which includes purpose as one of its six key components of wellbeing. They found that family connection predicted psychological wellbeing in adulthood across varying levels of childhood adversity. Finally, none of these studies investigated loneliness as a mediator. However, these studies provide promising evidence that the childhood family environment may predict meaning in life in adulthood and that this link may be explained by self-reported levels of adult loneliness.

### 1.4. The Present Study

The current study investigates the connection between meaning and the childhood family environment by addressing the following research questions: Does the childhood family environment affect emerging adults’ sense of meaning in life, such that emerging adults from emotionally warmer and less risky family environments report a greater sense of meaning in life? If so, is this connection mediated by loneliness—in other words, do emerging adults from warmer family environments report a higher sense of meaning in life because they are less lonely? If growing up in an emotionally warm, safe, and secure family environment translates into higher-quality relationships in adulthood and high-quality relationships are a source of meaning, then emerging adults from emotionally warm, safe, and secure family environments may also have a stronger sense of meaning in life.

Based on the existing research, one hypothesis of this study is that being from a warm, safe, and secure family environment predicts meaning in life, such that emerging adults from warmer, safer, and more secure family environments will report more meaning in life, and that this connection is mediated by loneliness. Given that prior research has shown that the childhood family environment predicts loneliness and that loneliness predicts a lack of meaning in life, this study also hypothesizes that the relationship between the childhood family environment and meaning in life goes through loneliness. In other words, this study is seeking to test the theory that the reason why emerging adults from warmer, safer, and more secure family environments report more meaning in life is because they are less lonely.

## 2. Method

### 2.1. Participants and Procedure

The participants were 507 undergraduates (68% female) at a private urban Catholic university receiving partial course credit for participating in a larger study investigating development among emerging adults. Participants ranged in age from 18 to 24 years (*M* = 18.95, *SD* = 1.18); 67 percent were first-year undergraduate students. Eighty-six percent of participants reported their race as White; twenty percent reported being Hispanic. Given that religiosity is a predictor of meaning in life [58] and that this sample was drawn from a religious university, religiosity was controlled for in order to isolate the effects of the childhood family environment on meaning over and above religiosity. The participants anonymously completed a battery of measures online. Data were collected between Fall 2020 and Spring 2023. This study was approved by the university’s institutional review board.

### 2.2. Measures

Participants completed a battery of questionnaires, including measures of the early family environment, loneliness, and meaning in life. In the current study, the term childhood family environment refers to the presence of parental warmth (e.g., affection, support, nurturance, acceptance [33]) and the absence of dysfunction and risk in the family (e.g., marital conflict, substance abuse, violence [32]) from infancy through adolescence (0–18 years old).

#### 2.2.1. Childhood Family Environment

Two retrospective assessments were used to assess the early family environment.

The Modified Parental Bonding Instrument (PBI; [59]) is a retrospective self-report of two components of the parent–child relationship: demonstrations of care by the parent and parental overprotection. Participants indicate on a four-point scale, ranging from 1 (very like) to 4 (very unlike), the degree to which 12 items represent their experiences with their own mother or father while growing up. Sample items include: “Spoke to me with a warm and friendly voice”, “Invaded my privacy” (reverse scored), and “Seemed to understand what I wanted or needed”. Participants in this study filled out the PBI twice when applicable, one for their mother and one for their father. The measures showed good reliability in this sample (PBI-mother, Cronbach α = 0.86; PBI-father, Cronbach α = 0.86).

The Risky Families Questionnaire (RFQ; [32]) assesses respondents’ experiences of conflict, coldness, and neglect in the family environment during childhood. Participants indicate on a 4-point Likert-type scale ranging from 1 (rarely or none of the time) to 4 (most of the time) the degree to which 11 items applied to them growing up. Sample items include: “How often did a parent or other adult in the household swear at you, insult you, put you down, or act in a way that made you feel threatened?”, “In your childhood, did you live with anyone who was a problem drinker or alcoholic, or who used street drugs?”, and “How often would you say there was quarreling, arguing, or shouting between your parents?” The measure showed good reliability in this sample (Cronbach α = 0.86).

The PBI and RFQ were standardized, reverse-scored for ease of interpretation, and combined to create a composite score reflecting the childhood family environment. Because the PBI was assessed for mothers and fathers separately, PBI-mother and PBI-father were first averaged into a “PBI-total” score. The PBI-total and RFQ score for each participant were then standardized and averaged to create a composite score. Importantly, the correlations between PBI-mother (*M* = 28.02, *SD* = 6.59), PBI-father (*M* = 28.99, *SD* = 6.56), and RF (*M* = 16.51, *SD* = 5.42) justified combining the measures. The correlation between PBI-mother and PBI-father was *r* = 0.48, *p* < 0.001. The correlation between PBI-total and PBI-mother was *r*(449) = 0.49, *p* < 0.001. The correlation between PBI-total and PBI-father was *r*(454)= 0.86, *p* < 0.001. The correlation between RFQ and PBI-total was *r*(379) = 0.66, *p* < 0.001. High scores for the childhood family environment composite reflect emotionally warm and less risky family environments.

#### 2.2.2. Loneliness

The three-item Loneliness Scale [60] is used to measure of the degree of loneliness typically felt by participants. The short form is based on the R-UCLA Loneliness Scale [61] and shows good convergent validity with the longer measure. Items on the short form include, “How often do you feel that you lack companionship?”, “How often do you feel left out?”, and “How often do you feel isolated from others?” The items are scored on a 3-point scale (1 = hardly ever; 2 = some of the time; 3 = often). The measure showed good reliability in this sample (Cronbach α = 0.80). High scores indicate more loneliness.

#### 2.2.3. Meaning in Life

Meaning in life was measured using the presence-of-meaning subscale of the meaning in life questionnaire [1]. The five-item presence-of-meaning subscale includes items such as “I understand my life’s meaning”, “My life has a clear sense of purpose”, and “My life has no clear purpose” (reverse-scored). The items are scored on a seven-point scale from 1 (absolutely untrue) to 7 (absolutely true). The subscale showed good psychometric properties in this sample (α = 0.78). High scores indicate more presence of meaning.

#### 2.2.4. Religiosity

Participants reported the extent to which they considered themselves to be a religious person on a scale from 1 (not at all) to 5 (very) (M = 3.14, SD = 1.24) and the degree to which they find strength and comfort in their religion on a scale from 1 (never) to 6 (many times a day) (M = 3.57, SD = 1.61). The two ratings were highly correlated, *r(358)* = 0.72, *p* < 0.001. They were standardized and averaged to create a religiosity score that was entered into the analyses as a control variable.

### 2.3. Statistical Analysis

Path analysis [62] was used to test the mediational hypothesis of this study. Childhood family environment was the independent variable; meaning in life (*M* = 20.04, *SD* = 6.73) was the dependent variable; and loneliness (*M* = 5.78, *SD* = 1.84) was the mediator variable. The Sobol test was used to test the mediation effect. To determine the amount of variance explained by each exogenous variable, linear regressions were used per the Baron and Kenny method (1986). The following regressions were in SPSS (Version 28, IBM, Chicago, IL, USA): (1) childhood family environment predicting loneliness; (2) childhood family environment predicting meaning in life to get the total effect of the early family environment on meaning; and (3) childhood family environment and loneliness predicting meaning in life. Religiosity was controlled in each step of the regression analysis.

The hypotheses of this study are that the childhood family environment predicts meaning in life and that this effect is mediated by loneliness. For this to be true, the childhood family environment must predict meaning in life; the early family environment must predict loneliness; loneliness must predict meaning in life; and the relationship between the early family environment and meaning in life should be smaller when loneliness is included in the model than when it is not, suggesting that some of the effect of the early family environment on meaning is through loneliness.

## 3. Results

A series of multiple regressions was used to test the mediational path analysis model [62]. Religiosity was controlled in each step (see Table 1).

In the first step, as expected, greater emotional warmth, safety, and security in the childhood family environment (the predictor) was related to a greater sense of meaning in life in emerging adulthood (the outcome). In the second step, as expected, a warmer, safer, and more secure family environment in childhood predicted less loneliness (the mediator) in emerging adulthood. In the third step, less lonely participants reported a greater sense of meaning in life.

To test mediation, the childhood family environment (the predictor) and loneliness (the mediator) were entered into the model simultaneously, while religiosity again was entered as a control variable. The effect of the childhood family environment on meaning in life was weaker when loneliness was included in the model than when it was not, suggesting partial mediation. To determine if the mediating role of loneliness was significant, the Sobel test was conducted using the Sobel test calculator at quantpsy.org. The Sobel test indicated that the mediation was significant (*z* = 5.08, *SE* = 0.14, *p* < 0.001). These results support the conclusion that loneliness partially mediates the relationship between the early family environment and meaning in life.

## 4. Discussion

The results of this study provide support for the hypothesis that an emotionally warm, safe, and secure childhood family environment predicts higher levels of meaning in life in emerging adulthood and that this relationship is mediated by loneliness. Thus, it may be that the reason, in part, why emerging adults from emotionally warm, safe, and secure family environments experience more meaning in life is because they are less lonely, and this lower level of loneliness predicts their greater levels of meaning in life.

This study makes a number of contributions to the existing literature. First, it ties developmental factors to the formation of meaning in life in emerging adulthood. Individuals do not suddenly develop an interest in meaning or a sense of meaning once they enter adulthood. In childhood, they are forming an understanding of the world [63] and developing working models of relationships [23] that they carry with them into later years. Those early life experiences, in turn, may affect constructs that influence their health and wellbeing, such as how lonely they may feel or how meaningful they believe their lives to be. Indeed, this study supports the notion that the quality of relationships in the childhood family environment predicts one’s degree of loneliness and meaning in life in emerging adulthood. Thus, this study helps researchers understand an important precursor to meaning in life, the childhood family environment. This link has been largely overlooked in the research but is in line with current understandings of meaning in life, which stress the importance of relationships to meaning. Second, this study proposes the mechanism through which childhood experiences influence emerging adult mental health: loneliness. When children grow up in households that are emotionally warm, safe, and secure, they grow into emerging adults who feel less lonely, and this predicts their meaning in life. By contrast, when children grow up in family environments that are cold, controlling, and abusive, they experience more loneliness, and this loneliness predicts lower levels of meaning in life as they enter adulthood. Finally, this study contributes to the literature by examining meaning in life as an outcome variable. To date, researchers have investigated the link between childhood experiences and mental health factors in adulthood. However, very few have investigated meaning in life as an outcome variable, even though meaning is an important indicator of mental health and is associated with many positive physical health outcomes [64].

By showing a link between development and meaning, this study might also inspire more research on the developmental factors influencing meaning in life and other constructs. To date, much positive psychology research has focused on discrete domains that contribute to one’s experience of meaning in life in adulthood (e.g., spirituality, work, and current relationships; see Snyder and colleagues [65]). While this research is helpful in unpacking what the causes, predictors, and correlates of meaning are in adulthood, it does not tell us whether developmental factors might be antecedents, predictors, and correlates of those causes. If meaning has developmental antecedents, some questions researchers may address include: Does development in early or middle adulthood influence how meaningful people rate their lives to be in later life? How does one’s sense of meaning change throughout the lifespan? Are there periods in life where relationships are more important for meaning, such as in emerging adulthood? Do developmental factors moderate or mediate the relationship between a construct of interest and meaning in life—for example, is the relationship between belonging and meaning in life moderated by religious upbringing? Researchers could extend the findings of this study by testing its research question in more diverse populations or at different stages of life. For example, would the model proposed by this study apply to LGBTQ samples or African American samples? Are relationships and loneliness predictors of meaning for middle-aged adults or older adults? Understanding the developmental factors that impact one’s sense of meaning in life across adulthood may contribute to more research, richer theories of personality development, and a deeper understanding of the developmental trajectory of meaning among various populations.

At the level of clinical practice, this study has several implications. For clinicians working with adults, it may be important to understand the link between development and meaning. If an adult’s emotionally cold, risky family environment has translated into fewer and lower quality relationships in adulthood, then one way to boost the client’s sense of meaning may be to help them strengthen their adult relationships. The clinician might also help a client develop or restore a sense of meaning in life with interventions targeted at building social support and belonging in different contexts. For these clients, it may be especially important for the therapeutic alliance to be strong and the dynamic between the therapist and client to be perceived by the client as warm, safe, and connected, allowing the client to see that such connections are possible and may be healing. Understanding the precursors and predictors of meaning may help clinicians create and implement interventions that can help individuals flourish.

This study suffers from several limitations. First, the participants in this study were undergraduates (mostly first-year students) at a private, religious institution and formed a convenience sample. Though religiosity was controlled in the analyses, caution should still be taken when generalizing to the broader population. Second, although the findings of this study support the proposed theoretical model, the cross-sectional nature of the data and the direction of effect cannot be known with certainty. A longitudinal study would allow for a more robust examination of the mediational model and allow for comparison to alternative interpretations. Notably, the mediating effect of loneliness was partial, which suggests that other constructs might also help explain the relationship between the childhood family environment and meaning. Future research might include more variables in the model and conduct a path analysis to see if more variance might be explained by the inclusion of other variables, such as self-compassion.

Finally, it is important to note that this study took place within the context of the global COVID-19 pandemic. Data collection spanned the autumn of 2020 to spring 2023. The pandemic may have had significant effects on many of the emerging adults in this sample. Some emerging adults may be suffering from higher rates of loneliness and a lack of meaning in life as a result of disrupted routines and social isolation. Indeed, research during the pandemic indicates that loneliness rose and meaning in life decreased among various adult samples [66,67,68]. However, several factors mitigate this limitation. First, the students at this specific university were mostly on campus for classes from fall 2020 to spring 2021, unlike their peers at other institutions; and the students in this sample were then fully on campus from spring 2021 to spring 2023. This may explain why this sample appears to report lower levels of loneliness compared to peers in other studies. Other samples of emerging adults generally reported scores in the lonely range during the pandemic [67], while the loneliness scores of this sample (*M* = 5.78, *SD* = 1.81) indicated that they were not lonely [60]. In addition, the students in this sample reported scores on the meaning in life questionnaire (*M* = 21.04, *SD* = 6.73) that were similar to same-aged samples in other research studies examining meaning in life among emerging adults before the pandemic [1]. These factors suggest that the present study may perhaps apply to the broader population of emerging adults post-COVID, though future research is needed to determine this.

## 5. Conclusions

In summary, this study found that participants who reported growing up in an emotionally warm, safe, and secure family environment ultimately reported more meaning in life as emerging adults and that this link was partially explained by their lower levels of loneliness. Overall, these findings suggest that people from emotionally cold and risky childhood family environments may struggle with meaning in life as emerging adults because they are lonelier. This research, though suffering from some limitations, nonetheless contributes to the body of knowledge on meaning in life by offering a developmental perspective.

## Figures and Tables

**Table 1 ijerph-20-05945-t001:** Regression Analysis for Mediation of Loneliness between Family Environment and Meaning, Controlling for Religiosity.

Variable	B	SE	t	F	R^2^
Step 1 (DV: Meaning)				76.117 ***	0.232
Intercept	21.070	0.263	80.233 ***		
Religiosity	2.570	0.278	9.235 ***		
Family Environment	2.019	0.286	7.070 ***		
Step 2 (DV: Loneliness)				37.779 ***	0.130
Intercept	5.782	0.075	76.977 ***		
Religiosity	−0.223	0.080	−2.803 **		
Family Environment	−0.639	0.081	−7.840 ***		
Step 3 (DV: Meaning)				79.468 ***	0.240
Intercept	27.355	0.885	30.907 ***		
Religiosity	2.474	0.278	8.886 ***		
Loneliness	−1.090	0.146	−7.460 ***		
Step 4 (DV: Meaning)				64.038 ***	0.276
Intercept	25.921	0.910	28.491 ***		
Religiosity	2.382	0.272	8.743 ***		
Family Environment	1.483	0.294	5.046 ***		
Loneliness	−0.839	0.151	−5.555 ***		

Note. *** *p* < 0.001, ** *p* = 0.005. DV: Dependent variable.

## Data Availability

For access to the data of this study, please contact dameron@cua.edu.

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
