# Peer review of "The Relationship between Meaning in Life and the Childhood Family Environment among Emerging Adults"

_ijerph, 2023, doi:10.3390/ijerph20115945_

Round 1
Reviewer 1 Report
This study is meaningful because it may elucidate the relationships between childhood family environment and meaning in emerging adulthood from a developmental perspective. However, given its major revision needed, please check an attached file.

Author Response
Dear Reviewer,
We appreciate the thoughtful comments you have provided. We are grateful for the opportunity to revise our manuscript. We have addressed below the comments provided by you below, and are confident the manuscript is stronger as a result. Thank you for the time and effort you put into our manuscript.
Yours,
Emily Dameron, M.A., and Marcie Goeke-Morey, Ph.D.
The Catholic University of America
Washington, D.C., United States
“When the author tried to set out the importance of the construct “meaning in life,” there are several detailed descriptions about relationship between meaning and physical health besides mental illness. Then the reader would expect that there may be further comments regarding physical health later, which was not.”
Because these analyses do not include measures of physical health, we deleted mention of physical health in the introduction and significantly shortened it in the discussion.
“Line 35: The author mentioned that understanding the precursors and predictors of meaning may help policymaker. But there is no implication for the policymaker in discussion at the end. It would be better to either elicit it or put more implication in discussion part.”
Reference to policy is removed from introduction.
“Line 212: What are the relevant references of ‘household dysfunction’ children experienced at home? How could author insist that assessing it besides ‘parental warmth’ is one of the most common ways to determine the health of the childhood family environment.”
We now include more description and references in the introduction of risk in the family environment so that readers may understand how it relates to the current study and measures.
“Line 297: How many items RFQ has?”
RFQ has 11 items. This has been added to the measures section.
“Line 326: I was wondering if this measure [meaning in life] was the best option for the author”
The Meaning in Life Questionnaire is one of the most widely used and cited measures of meaning in life within psychological research. We think it served our purpose for this project.
“From line 408 to 415, from line 422 to 428: Two paragraphs are almost identical. The author must correct this error.”
The redundant text has been deleted.
Reviewer 2 Report
Please see the attached file for comments

Author Response
Dear Reviewer,
We appreciate the thoughtful comments you have provided. We are grateful for the opportunity to revise our manuscript. We have addressed below the comments provided by you below, and are confident the manuscript is stronger as a result. Thank you for the time and effort you put into our manuscript.
Yours,
Emily Dameron, M.A., and Marcie Goeke-Morey, Ph.D.
The Catholic University of America
Washington, D.C., United States
“This is a correlative study, and the author cannot use causal terminology”
Thank you for this point. The author corrected this on line 222 by changing the word “affect” to “predict.” Language has been adjusted throughout.
“The authors proposed to treat meaning in life as the dependent variable and loneliness as the mediator… I propose … the family environment may predict … loneliness, while meaning in life may serve as a mediator.”
We believe our proposed model is supposed by theory and previous research. Unfortunately the cross sectional nature of the data does not allow for model comparison; we discuss this limitation in the discussion and call for future research to explore alternative models.
“In the instrument description, the family environment was presented as a positive measure …. It was combined with the Risky Families Questionnaire, yet the correlations between them was positive (?). In addition, the correlation between family environment scores and loneliness was positive, and negative for meaning of life. Please explain…. Similarly, see the negative B value in the regression analysis, “greater emotional love, support and security in the early family environment was related to a greater sense of meaning in life in emerging adulthood (b = -2.89)…”
In the Parental Bonding Inventory and the Risky Family Questionnaire, lower scores indicate warmer, less risky family environments; in the Meaning in Life Questionnaire higher scores reflect more meaning. Thus, warmer and less risky family environments (lower values) predict more meaning (higher values). We recognize that the resulting negative coefficient is confusing, so in this revision for ease of interpretation, we have reverse-scored the family environment composite so that higher scores now reflect a warmer, less risky family environment. The valence of the b values changed as a result, and hopefully the results are more intuitive now.
In the limitation section, the authors mentioned the COVID-19 period and the religious college. These are two important aspect that should be presented in the introduction, supported by the existing research about emerging adults and loneliness during the COVID-19, and the stay-at home policy impact on families. They also have to present studies on the role of the religion that has been proposed as a strong predictor of
meaning in life (Yaden at al., 2017).
Because of the nature of the sample, the authors controlled for religion in the analysis. Notably, the pattern of results remained the same. We did not include mention of COVID-19 in the because this study is not a COVID-19-specific study but we do address it in the discussion. Notably our data indicates that our sample of students did not report higher levels of loneliness during the pandemic, as did other samples of emerging adults.
Reviewer 3 Report
There is a need for more research that explores positive psychology constructs. The key role that development plays in forming adult behavior patterns is also appreciated.
Yet, the article could be significantly abbreviated. There are multiple instances where the paper repeats the same concept in nearly identical language two or more times within a single paragraph.
Additionally, there are multiple instances where concepts are described in detail where a minor information is needed. For example, the discussion of attachment theory could be signficantly abbreviated. The process for investigating mediation can also be removed, with a reference for those unfamiliar with the analysis.
Overall, this is a useful contribution to the field, but the paper needs to be more concise.
Author Response
Dear Reviewer,
We appreciate the thoughtful comments you have provided. We are grateful for the opportunity to revise our manuscript. We have addressed below the comments provided by you below, and are confident the manuscript is stronger as a result. Thank you for the time and effort you put into our manuscript.
Yours,
Emily Dameron, M.A., and Marcie Goeke-Morey, Ph.D.
The Catholic University of America
Washington, D.C., United States
“Yet, the article could be significantly abbreviated. There are multiple instances where the paper repeats the same concept in nearly identical language two or more times within a single paragraph.”
We have edited accordingly and removing repetitive material.
“Additionally, there are multiple instances where concepts are described in detail where a minor information is needed. For example, the discussion of attachment theory could be significantly abbreviated. The process for investigating mediation can also be removed, with a reference for those unfamiliar with the analysis.”
We shortened the descriptions of concepts such as attachment theory and the explanation of the mediation model.
Round 2
Reviewer 1 Report
As far as presenting a table of the result which helps the reader to easily figure out this study, overall I think that this manuscript can be accepted. Thank you for your hard work.
Author Response
Dear Reviewer 1,
Thank you for reviewing our manuscript once more. We have now included a table delineating the results of the our study. We hope that the results are now more clear as a result. Thank you for this suggestion.
Yours,
Emily Dameron, M.A., and Marcie Goeke-Morey, Ph.D.
Catholic University of America
Washington DC
Reviewer 2 Report
I don't have additional comments
Author Response
Thank you for reviewing our manuscript once more.